# Marginal Effects and Spatial Variations of the Impact of the Built Environment on Taxis’ Pollutant Emissions in Chengdu, China

**DOI:** 10.3390/ijerph192416962

**Published:** 2022-12-16

**Authors:** Guanwei Zhao, Zeyu Pan, Muzhuang Yang

**Affiliations:** School of Geography and Remote Sensing, Guangzhou University, Guangzhou 510006, China

**Keywords:** taxis’ emissions, built environment, marginal effect, spatial nonstationary, big data, spatiotemporal

## Abstract

Understanding the impact of the urban built environment on taxis’ emissions is crucial for sustainable transportation. However, the marginal effects and spatial heterogeneity of this impact is worth noting. To this end, we calculated the taxis’ emissions on weekdays and weekends in Chengdu, China, and investigated the impact of the built environment on taxis’ emissions by applying multi-source data and several spatial regression models. The results showed that the taxis’ daily emissions on weekdays were higher than the emissions on weekends. The time heterogeneity of hourly taxis’ emissions was not significant, while the spatial heterogeneity of taxis’ emissions was significant. Except the HHI, the selected built environment variables both had a significant positive effect on taxis’ emissions on the global scale. There was a marginal effect of some built environment variables on taxis’ emissions, such as the density of bus stops and population density. The former exhibited an inhibitory effect on taxis’ emissions only when it was greater than 9 stops/km^2^, while the latter showed an inhibitory effect only in the range between 16,000 people/km^2^ and 22,000 people/km^2^. There were some spatial variations in the effects of built environment factors on taxis’ emissions, with HHI, road density, and accommodation service facilities density showing the most significant variation. The marginal effect and spatial variation of the impact needs to be considered when developing strategies to reduce taxis’ pollutant emissions.

## 1. Introduction

With the rapid increase in greenhouse gas emissions, climate change has become an issue of global concern. China has made active efforts to reduce greenhouse gas emissions and has set “peak carbon dioxide emissions” and “carbon neutrality” targets. The number of motor vehicles in China has surged from 16.09 million in 2000 to 302 million in 2021 [1,2]. It goes without saying that the rapidly growing carbon emissions from urban transport are one of the great challenges for China to achieve peak carbon and carbon neutrality. Studies on motor vehicle emissions have received the attention of many scholars [3,4,5]. Nowadays, GPS devices that can record the real-time location of vehicles have been widely used in the transportation field. Since GPS data can reflect information such as vehicle travel speed, travel time, and occupant status, it helped us to analyze urban traffic characteristics dynamically from both time and space dimensions [6,7]. GPS data on taxis were widely used in research because of their easy access, various data contents, and high accuracy. The driving speed recorded by GPS trajectory can reflect the traffic condition of urban roads to a certain extent, and the trajectory data provide the possibility of the accurate and convenient measurement of traffic emissions. Therefore, the analysis of the spatiotemporal characteristics of traffic emissions through taxis’ trajectories has become one of the research priorities [8].

Transportation emissions reduction is one of the common issues in the global response to climate change, and it is also a way to achieve the sustainable development of cities and transportation. Scholars have explored the issue of transportation emissions mainly from three perspectives: national energy, urban form, and community scale. Previous studies at the national and city scales have yielded more consistent findings. In recent years, traffic emission studies have focused on the impact of the built environment on traffic emissions, but there are still some shortcomings. For example, differences or even opposite conclusions from different cases; is there a marginal effect of built environment factors on traffic emissions?

To this end, we investigated the marginal effects and spatial heterogeneity of the built environment’s effects on taxicab traffic emissions by applying the least squares regression (OLS) model, the geographically weighted regression (GWR) model, the random forest (RF) model, and the partial dependence plot (PDP). Consequently, the results will benefit our understanding of the mechanism of the built environment’s influence on traffic emissions and provide guidance for low-carbon-oriented built environment planning.

## 2. Literature Review

Traffic emissions are one of the classic topics of concern for many scholars. There are two main methods for measuring traffic emissions that are commonly used internationally. The first one is the “top-down” approach based on fuel, which calculates the emissions from energy consumption [9]. The second is a “bottom-up” approach based on the distance travelled to calculate the emissions generated during vehicle travel [10]. The fuel-based “top-down” approach is more suitable for national or regional emission measurements. For taxicab traffic emissions, the distance-based “bottom-up” emission calculation method is more reasonable. At present, the mainstream approach of taxi traffic emissions measurement is to extract the characteristics of actual vehicle driving conditions based on trajectory data and estimate them with the help of motor vehicle emission models. Existing motor vehicle emission models are mainly categorized into models based on average speed and models based on driving conditions, according to the simulation methods and required vehicle operating parameters [11]. The former mainly include COPERT (Computer Programme to Calculate Emissions from Road Transport), EMFAC (Emission Factors), and MOBILE (Mobile Source Emission Factor Model). The latter mainly include CMEM (Comprehensive Modal Emission Model), IVE (International Vehicle Emission Model), and MOVES (Motor Vehicle Emission Simulator). According to a comparison study proposed by Jin et al. [12], the MOBILE model is more mature, highly applicable, and suitable for emission measurement at the medium and macro levels; the COPERT model is less perfect than the MOBILE model and more suitable for Europe and developing countries with a lack of data; the MOVES model has the most perfect structure and can measure different levels, but it is not easy to apply locally; the IVE model has strong micro-simulation capability, good applicability, and high accuracy, but the model application is complicated; and the CMEM model has strong usability and high accuracy, but has high requirements for the quality of underlying data. In short, the decision of which emission model to use should be based on different application scenarios.

Many scholars have applied vehicle emission models to estimate traffic emissions. For example, Luo et al. (2017) analyzed the spatiotemporal characteristics of taxis’ pollutant emissions in Shanghai, China using taxis’ GPS data [13]. Shang et al. (2014) inferred vehicle pollutant emissions in Beijing, China by integrating taxis’ trajectory data as well as geographic information data [14]. Zeng et al. (2007) identified the air pollutant emissions of taxis based on their trajectory data and the artificial neural network approach [15]. Nyhan et al. (2016) predicted the spatiotemporal distribution of traffic pollutant emissions in Singapore by utilizing taxis’ GPS data and a microscopic carbon emission model [4]. Liu et al. (2019) reconstructed the spatiotemporal distribution of urban vehicle pollutant emissions by using multi-source data such as taxis’ GPS data, license plate recognition data, and geographic information data [16]. In short, numerous studies have proved that taxis’ GPS data can effectively promote the transportation emission measurement by time and provide auxiliary support for the formulation of urban emission reduction policies.

The relationship between traffic emissions and the urban built environment has received the attention of scholars both at home and abroad. According to the five Ds theory proposed by Ewing et al. [17,18], built environment elements usually include density, design, diversity, distance to transit stops, and destination accessibility. Specific evaluation indicators include population density, land-use mixture, road network density, bus-stop density, distance to bus stops, distance to downtown, residential density, and metro station density [17,19,20]. Some studies have pointed out that population density, land-use mixture, road network density, and bus-stop density have a suppressive effect on traffic emissions [21,22,23]. For example, Shim et al. (2006) stated that urban population size, population density, and road density had a negative effect on urban traffic emissions [24]. Barla et al. (2011) claimed that traffic emissions were 27% higher in less densely populated suburban areas than in more densely populated urban centers in Quebec, Canada [25]. Chai et al. (2011) found that land-use mixture and metro accessibility had a significant effect on traffic emission reduction in Beijing, China and general public transport accessibility had a weak contribution to traffic emissions [26]. Modarres et al. (2013) found that residential commuting traffic emissions were lower in areas with high population density in Los Angeles, USA [27]. Huang et al. (2014) noted that land-use mixture, road network density, and bus routes density had a suppressive effect on residential travel emissions at the community scale in Wuhan, China [28,29]. Zhang et al. (2020) found that the average service area per school also had a significant negative effect on traffic emissions [30]. A case study by Wu et al. (2019) for the Minneapolis St. Paul metropolitan area showed that distance to the nearest transit stop, employment density, and land-use diversity had a significant effect on traffic emissions [19]. Among the built environment indicators, the distance of residential areas from work areas (job-dwelling distance) is also a significant factor influencing traffic emissions. For example, Tong et al. (2012) concluded that reducing the distance between work and residence was significant for reducing traffic emissions [31]. The imbalance in distance between residential and work areas has led to an increase in traffic emissions due to increased long-distance commuting [32]. It can be seen that a lot of research has been conducted on the relationship between the urban built environment and traffic emissions. However, the findings also varied considerably from region to region [33,34]. It is generally accepted that traffic emissions are significantly greater in densely populated settlements than in other settlements [35]. However, Brand et al. (2013) found that the effect of population density on traffic emissions was relatively weak in UK regions [34]. Most studies concluded that metro accessibility has a significant inhibitory effect on traffic emissions, while general public transportation has no significant effect on traffic emission reduction [36,37]. And yet, some studies have found that general public transportation has a contributing effect on traffic emissions in Beijing, Guangzhou, and Shanghai [38,39,40]. In addition, some scholars have even reached diametrically opposed conclusions regarding the impact of the built environment of residence and workplace on traffic emissions [41,42]. Zhu et al. (2019) found a nonlinear relationship between the urban built environment and traffic emissions, and argued that the effect relationship only played a significant role in a specific range [20]. In summary, the mechanism of the built environment’s influence on traffic emissions varies considerably among the cases in different study areas. In other words, the differences and even contradictions between findings proposed by scholars were non-negligible.

Therefore, the following questions still urgently need to be answered. For example, is the relationship between the urban built environment and taxicab emissions non-linear? What is the marginal effect of the impact of the urban built environment on taxicab emissions? How does the influence of the urban built environment on taxicab emissions vary spatially? Answering the above questions can help achieve more efficient and accurate results in low-carbon transportation practices.

## 3. Materials and Methods

### 3.1. Study Area

Chengdu city is the capital of Sichuan Province, China. The total area of Chengdu city is 14,335 square kilometers. The eastern region of Chengdu is the hinterland of the Chengdu Plain, with elevations generally above and below 750 m and the lowest elevation at 359 m. In 2014, the number of passenger trips by taxis in Chengdu was 253.25 million, the number of passengers carried was 429.62 million, and the mileage operated was 221.505 million kms [43]. The taxis’ trajectory data involved in this paper were mainly distributed within the area of the Chengdu Belt Expressway (also known as the Fourth Ring Road). The location of the study area is shown in Figure 1.

### 3.2. Data Source and Preprocessing

The data sources are shown in Table 1. The taxis’ GPS record data were provided by the Intelligent China Cup, 2016 (https://challenge.datacastle.cn/v3/cmptDetail.html?id=175, accessed on 12 November 2016). The fields of data include vehicle ID, latitude, longitude, passenger load, date, and time. The raw data provided by the competition included more than 1.4 billion GPS records of 14 million taxis in Chengdu, from 3 August 2014 to 30 August 2014. Duplicate and abnormal records were removed and the records in the period 00:00:00~05:59:59 were dropped by the competition organization. The coordinate system of the longitude and latitude fields is the WGS1984 coordinate system. In this paper, the taxis’ trajectory data of 3 August 2014 and 4 August 2014 were selected for taxis’ pollutant emissions calculation, which represent the characteristics on weekends and weekdays, respectively. Among them, 3 August 2014 is a Sunday and 4 August 2014 is a Monday. The weather of Chengdu city on these two days is no rainfall and will not affect the taxi-travel behavior of residents. Since the number of taxi trips is very low between 0:00 and 6:00, the results measured in this paper can be considered approximately equal to the emissions of the whole day. The administrative boundary vector data were collected from the road traffic monitoring platform of Chengdu. The collection date was June 2021, the data format is geojson, and the coordinate system is the WGS1984 coordinate system. The POI data were obtained programmatically using the AMap Web Service API (https://lbs.amap.com/, accessed on 7 March 2018), which includes fourteen POI categories: catering facilities; scenic spots; public service facilities; companies; shopping facilities; transportation facilities; financial facilities; educational, scientific, and cultural facilities; residence district; living service facilities; sports and leisure facilities; medical service facilities; government agencies; and accommodation service facilities. Duplicate and abnormal records were removed and a total of 274,175 records were left for subsequent processing. The 100 m resolution population data were downloaded from the website of WorldPop project (https://www.worldpop.org/, accessed on 17 July 2021). The bus station data were extracted from the POIs. Finally, the coordinate systems of all spatial data were unified as the WGS1984 UTM Zone 48N coordinate system.

### 3.3. Methods

The technical flowchart of our study is shown as Figure 2.

#### 3.3.1. The Setting of Traffic Analysis Zone (TAZ) Unit

The common traffic analysis units mainly include regular grid and road network units. The 1000 m regular grid was chosen as the traffic analysis unit in our study. Compared with the traffic analysis units obtained by using the road network, the regular grid can reveal the spatial variation of transportation emission patterns more clearly. Moreover, the number of traffic analysis units is larger, and the sample size is more adequate for regression analysis. We created a 1000 m square grid using the create fishnet tool of ArcGIS software (Esri, Redlands). Moreover, we used the toolsets in ArcGIS software (Esri, Redlands) such as zoning statistics, spatial join, and field calculator to calculate the built environment indicator values for each TAZ unit.

#### 3.3.2. The Calculation Method for Taxis’ Traffic Emissions

First, the taxis’ GPS data were connected into the trajectory line of each vehicle according to the vehicle license plate and time order. Then, the traffic emissions of trajectories were calculated using a motor vehicle emission model. As mentioned above, the taxi GPS data used in this paper were collected in August 2014. At that time, China’s motor vehicle emission standard was the China’s Stage IV emission standard (National IV standard), which was almost the same as the European emission standard [44]. Therefore, the COPERT model [45], which is applicable to the European emission standards, was applied to calculate taxis’ traffic emissions in our study. In addition, many scholars have applied the COPERT model to the study of urban motor vehicle emissions in China, confirming the high accuracy of the model [40,46]. The COPERT model categorizes motor vehicle emissions into thermal emissions and cold-start emissions according to the operating state of the engine [47]. Since the cold-start emissions accounted for a relatively small percentage, only the hot emissions of taxis were considered in this paper. The COPERT model calculates the hot emissions for each trajectory segment based on the distance and emission factors [47]. The emissions of each trajectory line segment were calculated as follows:(1)$$E_{j}^{k}=C_{j}^{k}\times l_{j}$$
where $E_{j}^{k}$ denotes the emission of pollutant k on trajectory line segment j; $l_{j}$ is the distance traveled by the vehicle on trajectory line segment j; and $C_{j}^{k}$ is the emission factor of pollutant k on trajectory line segment j, which is usually expressed as g·km^−1^. In the COPERT model, the emission factor is closely related to the average speed, and its general formula is shown as follows:(2)$$C_{j}^{k}=\frac{a_{k}\times {v_{j}}^{2}+b_{k}\times v_{j}+c_{k}+d_{k}/v_{j}}{e_{k}\times {v_{j}}^{2}+f_{k}\times v_{j}+g_{k}}$$
where $v_{j}$ is the average speed (km·h^−1^) of a motor vehicle on trajectory line segment j, and $a_{k}$, $b_{k}$, $c_{k}$, $d_{k}$, $e_{k}$, $f_{k}$, and $g_{k}$ are model parameters determined by vehicle type, emission standard, fuel type and engine type, respectively. The COPERT model has calibrated these parameters based on experimental data. Since the vast majority of taxis in our study were gasoline vehicles that meet the limits of China’s Stage IV emission standard, the vehicle type in the COPERT model was set to be a small passenger car, the fuel type to be gasoline, and the emission standard to be Euro IV (comparable to China’s National IV standard). The emissions analyzed in our paper include carbon dioxide (CO_2_), carbon monoxide (CO), nitrogen oxides (NO_X_), and hydrocarbons (HC). The detailed model parameters are shown in Table 2.

#### 3.3.3. Evaluation Indicators of the Built Environment

The popular “5Ds” model, which was proposed by Ewing and Cervero in 2010 [18], was applied to evaluate the urban built environment. The 5Ds theory measures the urban built environment from five aspects: design, diversity, density, distance to transit, and destination accessibility. The specific indicators system is shown in Table 3.

Except for the land-use mixture, the remaining indicators were obtained using the number divided by the area of the TAZ unit. The land-use mixture was measured by the Herfindahl Hirschman Index [48], which can be calculated as follows:(3)HHIi=∑j=1k(AijAi)2,
where $HHI_{i}$ denotes the land-use mixture of unit i, $A_{i}$ is the number of POIs in unit i, $A_{ij}$ is the amount of POIs type j in unit i, and k is the categories of POIs. For ease of description, a simplified name was given for each metric (see Table 3).

#### 3.3.4. The Global Regression Model for the Impact of the Built Environment on Taxis’ Emissions

The OLS model is usually the initial step of regression between the independent and dependent variables. The first step in the use of OLS models is the multicollinearity test of the independent variables. The correlation coefficient was utilized to perform a multicollinearity test with a threshold value of 0.8. The OLS regression model can be described as follows:(4)$$y_{i}=\sum_{i=1}^{n}\beta_{i}x_{i}+\varepsilon_{i},$$
where $y_{i}$ is the value of the dependent variable; $x_{i}\left(i=1,\;2,\;\ldots ,\;n\right)$ is the value of the independent variable; $\beta_{i}\left(i=1,\;2,\;\ldots ,\;n\right)$ is the coefficient of regression model; and $\varepsilon_{i}$ is the error term of the model. The OLS model is a global regression model, which is usually used to identify the significant built environment variables. In our study, the Python module named statsmodels (https://www.statsmodels.org/stable/index.html, accessed on 16 January 2022) was applied to perform the OLS regression analysis. Compared with weekdays, people’s travel behavior is more random during weekends, and the corresponding spatiotemporal characteristics of taxis’ traffic emissions are more unpredictable. Therefore, we chose the taxis’ emissions during weekends as the dependent variable and performed spatial regression analysis (including the global and local regression models) in our study.

#### 3.3.5. The Marginal Effect Model for the Impact of the Built Environment on Taxis’ Emissions

The marginal effect is derived from economics and refers to the new output or benefit from successively increasing the input of a factor when other inputs remain constant. In economics, there is generally a law of diminishing marginal utility, which means that the marginal utility decreases as the input is continuously increased. Marginal effects respond to nonlinear relationships between variables and are found in a wide range of fields. In our study, we applied the random forest (RF) algorithm to explore the nonlinear relationship between the built environment and taxis’ traffic emissions, and analyzed the marginal effects of the built environment’s impact on traffic emissions using partial dependency plots (PDP). The PDP is a global approach to inferring relationships between independent and dependent variables using the dataset. The PDP shows whether the relationship between labels and features is linear, monotonic, or more complex. The Python machine learning module Scikit-learn (https://scikit-learn.org/stable/, accessed on 12 April 2022) was applied to perform RF regression analysis and illustrate partial dependency plots.

#### 3.3.6. The Local Regression Model for the Impact of the Built Environment on Taxis’ Emissions

As we know, the value of the regression coefficient estimated by the OLS model is the average value of the entire study area, which cannot reflect spatial variation in the regression parameters. Spatial variation will lead to the spatial nonstationary relationship that affects the accuracy of regression results [49]. Therefore, the spatial nonstationary relationship needs to be addressed by applying local regression techniques such as the GWR model [50,51,52]. The basic GWR model can be expressed as follows:(5)$$y_{i}=\beta_{0}\left(u_{i},v_{i}\right)+\sum_{k=1}^{m}\beta_{k}\left(u_{i},v_{i}\right)x_{ik}+\varepsilon_{i},$$
where $y_{i}$ is the value of the dependent variable at position i; $x_{ik\;}\left(k=1,\;2,\;\ldots ,\;m\right)$ is the value of the independent variable at position i; $\left(u_{i},v_{i}\right)$ are the coordinates of position i; $\beta_{0}\left(u_{i},v_{i}\right)$ is the intercept term; $\beta_{k}\left(u_{i},v_{i}\right)\left(k=1,\;2,\;\ldots ,\;m\right)\;$ is the coefficient of regression model; and $\varepsilon_{i}$ is the error term of the model. The software developed by Oshan T.M et al. [53] (https://sgsup.asu.edu/sparc/multiscale-gwr, accessed on 8 February 2022) was applied to perform the GWR regression analysis. The spatial kernel of GWR regression was Fixed Gaussian, bandwidth search criterion was Golden section, the model type was Gaussian model, and the optimization criterion was AICc. The map of GWR coefficients was illustrated using the GeoPandas library, which was an open-source project to make working with geospatial data in python easier (https://geopandas.org/en/stable/).

#### 3.3.7. Evaluation Metrics of Regression Model

The adjusted R^2^ (adj. R^2^), residual sum of squares (RSS), Akaike information criterion, and corrected (AICc) [54] were applied to evaluate the regression results. The adj. R^2^ can be interpreted as the proportion of the variance of the dependent variable covered by the regression model. The closer its value is to 1, the better fitting the performance of the model. The RSS measures the level of variance of the error term or residuals of the regression model. The lower the RSS value, the better the regression model is at fitting the observed data. The AICc can be used to measure the practicality and complexity of the model. The AICc is not an absolute measure of goodness of fit, but is useful for comparing models that apply to the same dependent variable and have different explanatory variables. Once the difference between the AICc values of two models is greater than 3, the model with the lower AICc value will be considered the better model [52].

## 4. Results and Discussion

### 4.1. Spatiotemporal Characteristics of Taxis’ Pollutant Emissions

The total taxis’ emissions between 06:00 am and 23:00 pm on weekdays were 193,729.82 kg, of which CO pollutant emissions were 123.53 kg, NO_X_ pollutant emissions were 51.40 kg, HC pollutant emissions were 8.37 kg, and CO_2_ pollutant emissions were 193,546.52 kg. The total taxis’ emissions between 06:00 am and 23:00 pm on weekends were 172,721.87 kg, of which CO pollutant emissions were 117.06 kg, NO_X_ pollutant emissions were 44.86 kg, HC pollutant emissions were 7.72 kg, and CO_2_ pollutant emissions were 172,552.22 kg. It can be seen that the emissions of four pollutants were greater during weekdays than during weekends. We speculated that the reason may be mainly due to the more random travel behavior and higher average taxi travel speed during weekends. As we know, compared to weekdays, residents’ travel behavior on weekends is not constrained by commuting, the destinations of travel are more diverse, and the distribution of travel behavior over time is characterized by more randomness. Of course, the results in this paper are only based on sample data collected in two days, which may have the problem of small sample size. However, our finding about the temporal characteristic of taxis’ pollutant emissions is similar with a study performed by Liu et al. over a period of 9 days (22–30 June 2015) in Hangzhou, China [16]. The estimated hourly pollutant emissions from taxis during weekends and weekdays (represented by 3 and 4 August 2014, respectively) are shown in Figure 3.

As can be seen from Figure 3, taxis’ emissions exhibited very similar characteristics on weekdays and weekends. In order to avoid redundancy, the time characteristics of emissions will be analyzed using weekends as an example. First, emissions from taxis increased rapidly between 06:00 am and 10:00 am, peaking at 10:00 am. Then, it decreased between 10:00 am and 12:00 am. Moreover, emissions remained relatively stable between 13:00 pm and 18:00 pm, with a peak at 16:00 pm during this period. Finally, emissions continued to rise slightly between 19:00 pm and 22:00 pm, and then dropped sharply at 23:00 pm. To facilitate the analysis, 06:00 am to 23:00 pm was divided into three time periods: morning (06:00 am to 12:00 am), afternoon (12:00 am to 18:00 pm), and evening (18 pm to 23:00 pm). The emissions in the three periods were calculated and the resulting violin diagrams are illustrated in Figure 4.

As can be seen from Figure 4, there were similarities and differences in the statistical distribution of emissions in the three time periods during weekends and weekdays. The total taxis’ pollutant emissions of more than 50% of TAZ units in the study area were less than 100 kg and more than 75% of TAZ units were less than 700 kg. Among the three time periods, the emission distribution was the most discrete in the afternoon period and the emission peak value was the largest among the three time periods. During the weekend, the peak emissions in the morning period were the smallest among the three time periods. During weekdays, the peak emissions during the evening period were the smallest among the three time periods. Taxis’ pollutant emissions were significantly lower during weekend mornings compared to weekdays. These results revealed the characteristics of taxi traffic behavior of Chengdu residents at different time periods during weekdays and weekends. In addition, these results are similar with Zhang’s study in Beijing, China. In detail, Zhang et al. found that the spatiotemporal carbon emissions and travel patterns in Beijing, China differ between weekdays and weekends, especially during morning rush hours [39]. In order to further reveal the differences in taxis’ pollutant emissions throughout the day for each TAZ unit, the spatial distribution of taxis emissions on weekday and weekend are illustrated in Figure 5.

As can be seen from Figure 5, the emissions of four pollutants in the study area exhibited a similar distribution. The high-emission areas were mainly concentrated in the proximity of the Chengdu second Ring Road and the Chengdu second Elevated Ring Road, and Shudu Avenue. Among them, the taxis’ pollutant emissions in the areas including Fuqing Road interchange, the third section of the east of the second Ring Road, the intersection of Yusha Road and a section of Hongxing Road, and the intersection of the third section of the west of the first Ring Road and Yingmen Road were relatively high. The actual traffic conditions in the study area showed that the Chengdu second Ring Road and the Chengdu second Elevated Ring Road, as an important traffic ring road in Chengdu city, has a very high daily traffic flow, with congestion in the morning and evening peaks being more serious. There are a large number of residential areas, schools, hospitals, and traffic hubs along the Chengdu second Ring Road and the Chengdu second Elevated Ring Road, such as the Chengdu Railway Station and Southwest Jiaotong University (Jiuli Campus) in the northern section of the second Ring Road, and the Chengdu Sixth People’s Hospital and the University of Electronic Science and Technology (Shahe Campus) in the eastern section of the second Ring Road. The taxi flow in these locations is usually great, resulting in higher taxis’ pollutant emissions in this section. The pattern of higher taxis’ pollutant emissions is similar with the pattern in the research by Han et al. [55]. Han et al. illustrated the pattern of online car-hailing pollutant emissions in Chengdu, China in a 500 m grid scale by using the Didi on online car-hailing trip data in 2016. This result indicates a high degree of similarity between the cold hotspot areas for traditional taxi and online car-hailing trips in Chengdu. Shudu Avenue, which runs through the central city of Chengdu, has a large number of commercial centers and tourist attractions along its route, including Tianfu Square, Sichuan Science and Technology Museum, MaoYe Department Store, JinGuancheng Department Store, Chengdu Museum, and Chunxi Road. These commercial centers and tourist attractions attracted a large number of tourists and local residents, resulting in the increase of taxi pollutant emissions near Shudu Avenue. Some sections of intersections (interchanges), which act as important nodes connecting traffic flow in all directions, will inevitably bring huge taxi pollutant emissions in this area. In terms of the overall pattern, the spatial heterogeneity of taxis’ pollutant emissions in the study area was relatively significant.

### 4.2. Global Impact of the Built Environment on Taxis’ Emissions and Its Marginal Effect

First, the initial independent variables with a correlation coefficient greater than 0.8 were excluded. The remaining variables for modelling were DEN_bus_, HHI, DEN_pop_, DEN_road_, DEN_com_, DEN_sce_, DEN_acc_, and DEN_med_. Then, eight variables were subjected to OLS regression analysis. The adjusted R^2^ for OLS regression model was 0.673, indicating that the goodness of fit was satisfactory. The regression results (see Table 4) showed that the effects of these eight variables were significant, at least in the 0.05 level. The AICc value and RSS value of the regression model were 1023.397 and 189.551, respectively. This result is consistent with some previous studies. For example, Wu et al. found that three built environment factors have the strongest influences on CO_2_ emissions in the Minneapolis-St Paul twin cities area: distance to the nearest transit stop, job density, and land-use diversity [19]. However, there are also inconsistencies. For example, Yang et al. indicated that for different trip purposes, the effects of built environment elements on travel-related CO_2_ emissions were not consistent in Guangzhou, China. In particular, the role of distance to city public centers, residential density, and bus-stop density for commuting trips were likely to be different from trips taken for other purposes [33]. The detailed regression results are shown in Table 4.

The RF regression analysis was performed using eight significant variables to obtain importance rankings of independent variables and plot PDPs. First, all sample data were divided into a training set and a test set in the ratio of 9:1. Then, the GridSearchCV function was applied to the training set to obtain the best model estimation. The specific parameters of the best model estimation were as follows: number of estimators was 700, max_depth was 10, oob_score was false, and bootstrap was true. Finally, the best model estimation was applied to the sample set for regression analysis. The adjusted R^2^ of the regression result was 0.884, which was significantly higher than the result of the OLS model. The ranking of the importance of eight independent variables was consistent with the ranking of coefficients (absolute values) in the OLS model results. The PDP diagrams of the effects of eight built environment indicators on traffic emissions are shown in Figure 6.

The subplots in Figure 6 represent the changes in taxis’ emissions when only one variable is changed while holding the other variables constant. From Figure 6, it can be seen that there was a non-linear relationship between built environment factors and taxis’ emissions. This finding is consistent with some previous studies, such as that of Wu et al. which claimed that people should examine threshold effects of built environment elements on travel-related carbon dioxide emissions in the Minneapolis-St Paul Twin Cities area [19]. In detail, the densities of bus stops, companies, accommodation facilities, and the population have both promotion and inhibitory effects on traffic emissions. When the density of bus stops was less than 9 stations per square kilometer, taxis’ traffic emissions showed a rapid increase. The taxis’ pollutant emissions dropped slightly when the density of bus stops was greater than 9 stations per square kilometer. We speculated that mitigation of taxis’ pollutant emissions would only be possible if the density of bus stops was more than 9 stops per square kilometer. This pattern is not consistent with Wu’s study in the Minneapolis-St Paul twin cities area. In detail, Wu et al. found that when there are fewer than 20 stops within a half-mile buffer of a residence, the number of stops shows a weak association with CO_2_ emissions. Beyond this range, CO_2_ emissions begin to show a substantial decrease as the number of stops increases [19].

Taxis’ pollutant emissions showed a rapid decrease as the HHI was in the range from 0.2 to 0.3. However, taxis’ emissions remained almost constant when the HHI exceeded 0.3. Therefore, we believe that the aim of reducing taxis’ pollutant emissions cannot be achieved by simply pursuing an increase in land-use mixture. This negative relationship is consistent with many studies, such as from Wang [56]. However, the threshold of HHI in our study is different from the threshold (from 0.4 to 0.7) of HHI in the study by Wu et al. [19]. The inhibitory effect of population density on taxis’ pollutant emissions was mainly found in three density intervals. The most significant inhibitory effect was found in the range of 16,000 people/km^2^ to 22,000 people/km^2^. This pattern is not consistent with studies performed by Stevens and Wu [19,57]. For example, Wu et al. found that the effect of population density reaches a low point at about ten people/acre and then slowly increases [19]. Accordingly, we suggested that this marginal effect should be taken into account when adjusting the regional population density for taxi pollutant emission reduction purposes.

The taxis’ pollutant emissions remained almost constant as the road density was less than 5 km/km^2^, and it increased exponentially when the road density exceeds 5 km/km^2^. The slight inhibitory effect of company density on taxis’ pollutant emissions were only found in two intervals. The former was less than 20 companies per square kilometer, and the latter was from 55 companies/km^2^ to 75 companies/km^2^. A slight inhibitory effect of accommodation facilities density on taxis’ pollutant emissions was only found in the range of 18 per km^2^ to 28 per km^2^. In the range of sample values, the density of medical service facilities had a predominantly facilitative effect on taxi emissions. The growth rate of taxis’ pollutant emissions decreases significantly when the density of medical facilities exceeds 55 per square kilometer. In summary, the effects of the built environment indicators on taxis’ pollutant emissions were complex, and the marginal effects should not be ignored. The inconsistencies between our findings and other studies may be due to differences in study area, data type, nonlinear regression models, unit of measurement, and so on. For example, we applied the RF algorithm, while Wu et al. applied the gradient boosting decision tree algorithm. However, understanding the laws of the built environment’s effects on taxis’ pollutant emissions can provide more accurate references for local authorities to develop traffic emission reduction measures.

### 4.3. Spatial Variation in the Impact of the Urban Built Environment on Taxis’ Emissions

The diagnostic information of OLS model and GWR model is shown in Table 5.

As shown in Table 5, the residual distributions of OLS regression results have significant spatial autocorrelation, indicating the existence of spatial nonstationarity in the relationship between the urban built environment and taxis’ pollutant emissions. Therefore, the GWR model needs to be applied to deal with this spatial nonstationarity. In short, the fitting result of the GWR model was more reliable and robust than the result of the OLS model. The spatial kernel bandwidth of the GWR model results was 2217.87 m. The summary statistics for GWR parameter estimation results is shown in Table 6.

As can be seen from Table 6, the standard deviation of estimated coefficients for eight built environment indicators were higher than 0.1, besides the DEN_sce_. It can be inferred that the estimated coefficient values of indicators are statistically discrete. In other words, the spatial variations in the effect of the urban built environment on taxis’ pollutant emissions were significant. The indicator with the greatest degree of spatial variation in its effect was HHI, and the smallest was DEN_sce_. The estimated coefficients map of eight built environment indicators is shown in Figure 7.

The bus-stop density’s effect on taxis’ emissions showed an overall pattern that shifts from an inhibition effect in the central region to promotion effect in the peripheral region. In the central part of the study area and the areas along the northern part of the fourth ring road, the effect of bus-stop density on taxis’ pollutant emissions was negative. Due to the high density of bus stops in this region, there was a competitive relationship between bus and taxi that suppressed the traffic emissions from taxi trips. In the remaining areas along the fourth ring road, the effects of bus-stop density were mainly positive. It can be speculated that taxi travel is usually generated near the bus stations in this area, thus leading to an increase in taxis’ pollutant emissions.

Land-use mixture has a predominantly inhibitory effect on taxis’ pollutant emissions, showing a circling pattern with a gradual decrease in inhibitory effect from the inside out. Except for the area along the western part of the fourth ring road, the effect of population density on taxis’ pollutant emissions showed a general pattern of gradual increase from the inside to the outside. The promoting effect was most obvious in the southwestern part of the study area. The effect of road network density on taxis’ pollutant emissions showed an overall pattern of decreasing from the inside out. The promotion effect of road network density was most significant in the region within the second ring road. The area within the second ring road belongs to the downtown area of Chengdu, which has dense roads, convenient transportation facilities, and bustling business, thus bringing a strong demand for taxi trips.

The effect of company density on taxis’ pollutant emissions was both positive and negative. The promoting effect was mainly found in the central and southeastern parts of the study area, and the inhibiting effect was mainly found in the region along the western and eastern parts of the fourth ring road. To facilitate the analysis, we use company density to represent employment density in our study. The central and southeastern part of the study area is the preferred employment area due to its developed economy, high land-use mixture, and excellent transportation infrastructure. The increase in employment density in this region will attract more employed people, resulting in higher taxi pollutant emissions. In the southwest and east of the study area, the employment density was relatively low, and residents in this region may have long distances to commute. It can be speculated that there was some separation of jobs and residences in the study area.

The effect of scenic spots density on taxis’ pollutant emissions was both positive and negative. In the area along the southern and eastern parts of the fourth ring road, the public transportation is more developed than other regions. The transportation mode of tourists visiting these places tends to be public transport, thus leading to an inhibiting effect on taxis’ travel emissions. The transportation mode of tourists visiting the scenic spots in the western and southern regions of the study area tends to be private cars or taxis due to the insufficient public transportation facilities. Therefore, the higher density of scenic spots in this area will lead to the increase of pollutant emissions from taxi trips. Accommodation service facilities, which include hotels, apartments, and so on, serve mainly tourists of Chengdu. The most significant contribution of accommodation service facilities to taxis’ pollutant emissions was found in the north, northeast, and southwest of the study area. In addition, the area shape with the most significant promoting effect looked like the letter C. The higher density of accommodation service facilities in these areas will bring more temporary residential population, thus resulting in higher taxi pollutant emissions.

The effect of medical service facilities density on taxis’ emissions was also both positive and negative. The inhibitory effect was mainly found in the northern and southeastern regions, and the promoting effect was mainly found in the central and southern regions. The area shape with the most significant promoting effect was similar with the Greek alphabet gamma. We speculated that since high-grade hospitals in Chengdu were mainly located in these areas, they were more irreplaceable in the candidates of residents’ travel destinations [58], thus bringing higher taxi pollutant emissions. Our findings confirmed again that the spatial variations in the effect of the urban built environment on taxis’ pollutant emissions should not be ignored.

### 4.4. Policy Recommendations for Reducing Taxis’ Emissions

Based on the influence law of bus-stop density on taxis pollutant emissions, it is recommended to optimize the layout of bus routes and stops. Especially in the west and southeast of the study area, priority should be given to improving the density of bus stops.

According to the influence law of the density of employment areas on taxis’ pollutant emissions, we propose to enhance the proximity configuration of residential areas to employment areas, especially in the eastern and southeastern regions of the study area. By controlling the average employment-dwelling distance to a distance suitable for public transportation travel (5 km–15 km), the proportion of long-distance travel is reduced, thus reducing taxis’ pollutant emissions.According to the influence law of scenic spots density on taxis’ pollutant emissions, we recommend increasing the number of bus and metro stations in the eastern and southwestern regions of the study area to reduce the long-distance traffic emissions caused by citizens living in the peripheral areas who visit such places by taxi. According to previous research results, the metro lines can effectively reduce the pollutant emissions caused by taxis [59]. Therefore, we recommend that priority should be given to improving the coverage level of metro stations.The marginal effect of the impact of the built environment on taxis’ pollutant emissions needs to be considered when developing a low-carbon strategy. For example, the density of bus stops has a suppressive effect on taxis’ pollutant emissions only when it is greater than 9 stops/km^2^. Similarly, land-use mixture has a suppressive effect on taxis’ pollutant emissions only when it is lower than 0.3. Therefore, we argue that it may not be possible to reduce taxis’ pollutant emissions by simply seeking to increase the land-use mixture. The threshold of land-use mixture should not be ignored. In addition, the suppressive effect of population density on taxis’ pollutant emissions was only observed as the density value was from 16,000 person/km^2^ to 22,000 person/km^2^ in our study. Therefore, we suggested that the population density should be controlled within the range where the suppressive effect occurs when optimizing the regional population size.

## 5. Conclusions and Prospects

### 5.1. Conclusions

First, the total taxis’ pollutant emissions between 06:00 am and 23:00 pm on weekdays were 193,729.82 kg, while the total taxis’ pollutant emissions on weekends were 172,721.87 kg. Whether on weekdays or weekends, more than 75% of all TAZ units have less than 700 kg of taxi emissions. Among the three time periods, the taxi pollutant emission distribution was the most discrete in the afternoon time period and the emission peak was the largest among the three time periods. In terms of the overall pattern, the spatial heterogeneity of taxis’ pollutant emissions in the study area was relatively significant.

Second, the results of the global regression analysis showed that there were eight built environment variables that had a significant effect on the total pollutant emissions from taxis. The variables with positive effects in global scale were DEN_med_, DEN_acc_, DEN_pop_, DEN_road_, DEN_com_, DEN_sce_, and DEN_bus_ in descending order of magnitude. The variable with negative effects on the global scale was HHI.

Third, the results of the partial dependency analysis indicated that there was a marginal effect of some built environment variables on the total pollutant emissions from taxis. For example, the density of bus stops exhibited some inhibitory effects on taxis’ pollutant emissions when it was greater than 9 stops/km^2^. Population density has a suppressive effect on taxis’ pollutant emissions when it is in the range of 16,000 people/km^2^ to 22,000 people/km^2^.

Finally, the local regression analysis results revealed that there was a certain degree of spatial variation in the effects of the built environment on taxis’ pollutant emissions, with HHI, road density, and accommodation service facilities density showing the most significant variation characteristics. The effect of HHI on taxis’ pollutant emissions showed a circling pattern with a gradual decrease in inhibitory effect from the inside out. The promoting effect of road network density on taxis’ pollutant emissions showed a pattern of decreasing from the inside out. The promoting effect of accommodation service facilities density on taxis’ pollutant emissions was observed mainly in the north, northeast and southwest of the study area, and the inhibiting effect was observed in the eastern and southern parts of the study area.

### 5.2. Shortcomings and Prospects

First, the time of taxi GPS data used in this paper was relatively old, and some of the built environment indicator data did not coincide with the time of taxi GPS data. Our study ignored the possible changes in the indicator data during the period, which may lead to some errors in the results. In future studies, we should try to use the latest and consistent data sources to reveal the characteristics of urban traffic emissions more precisely.

Second, the impact of metro stations on taxi trips was not considered in the built environment indicators in our study. Urban rail transit is a green and low-carbon transportation mode, and has become a preferred transportation mode with its advantages of high punctuality, reduced delays, avoidance of ground congestion, and low travel costs. Since there is some competition between metro trips and taxis trips, it may have some impact on taxis traffic emissions. However, there was only one metro line (including 16 metro stations) in operation in Chengdu in August 2014. Therefore, it was not included in the built environment index system of our study.

Third, only traditional cruising taxis’ traffic emissions were investigated in this paper. However, urban public transportation systems are composed of various types, especially noting the new taxicab system represented by online-car hailing which has become mainstream. A comprehensive study on the traffic emissions of multiple public transportation types should be considered in the future, in order to develop more practical and feasible emission reduction measures.

## Figures and Tables

**Figure 1 ijerph-19-16962-f001:**
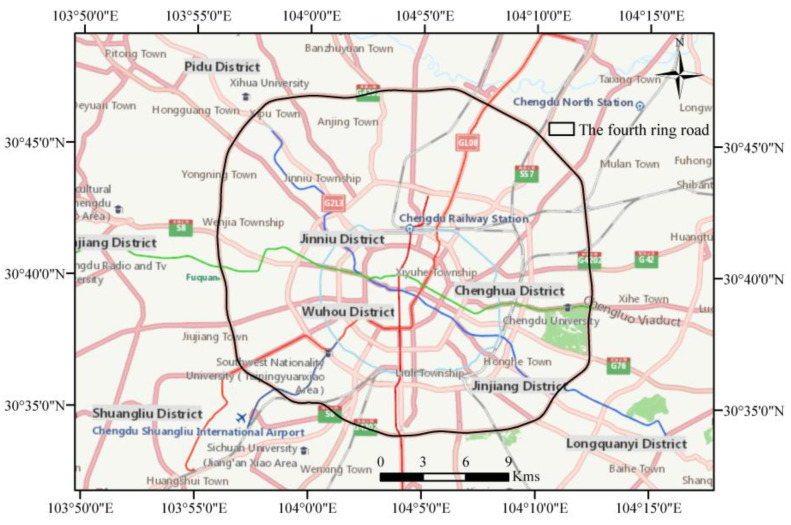
The study area map.

**Figure 2 ijerph-19-16962-f002:**
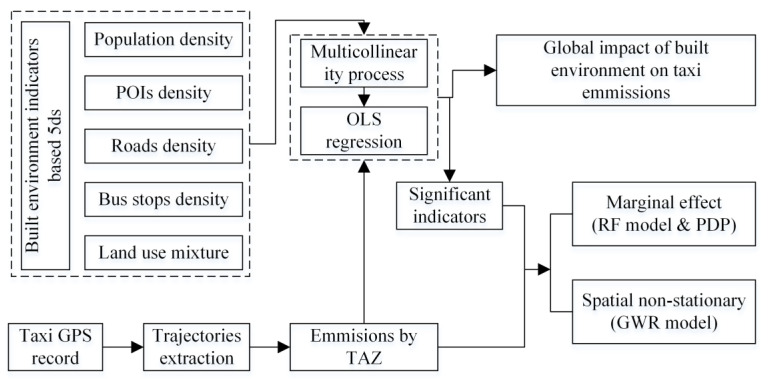
Flow chart of study.

**Figure 3 ijerph-19-16962-f003:**
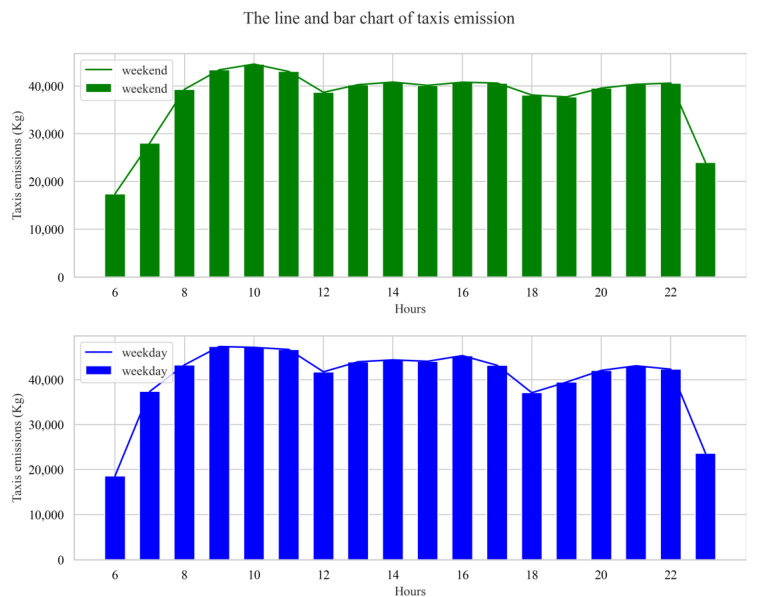
Hourly emission characteristics between 6:00 am and 23:00 pm.

**Figure 4 ijerph-19-16962-f004:**
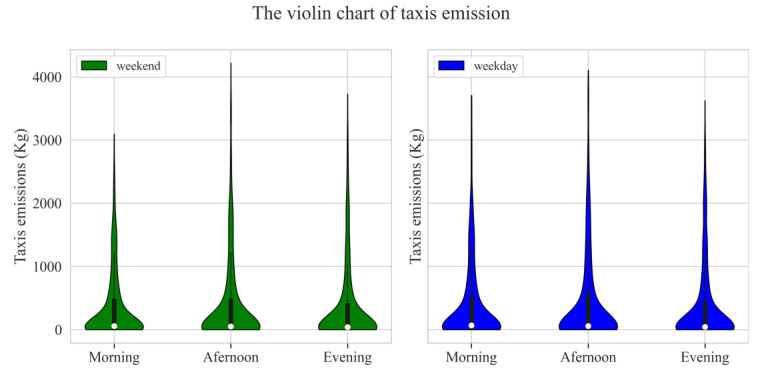
The violin diagram of taxis’ emissions.

**Figure 5 ijerph-19-16962-f005:**
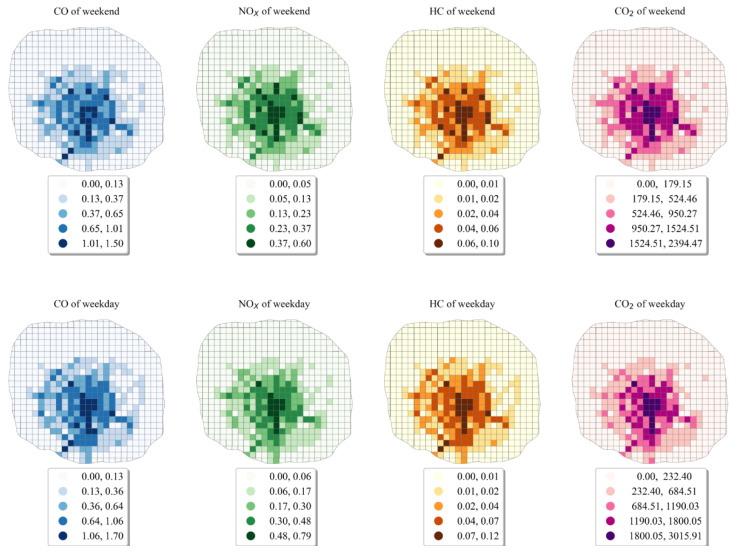
The emission distribution map (unit: kg).

**Figure 6 ijerph-19-16962-f006:**
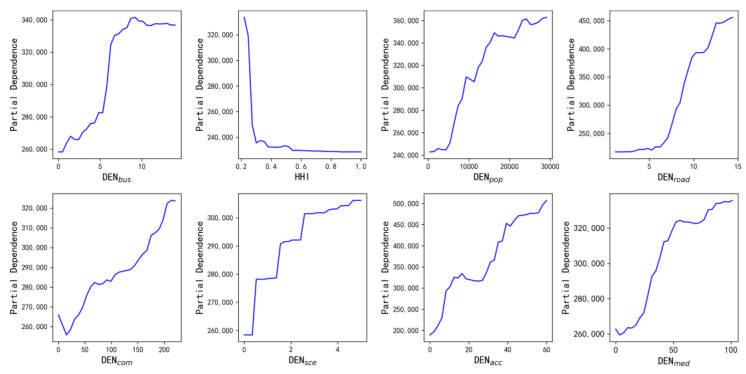
The PDP plots of built environment indicators on taxis emissions.

**Figure 7 ijerph-19-16962-f007:**
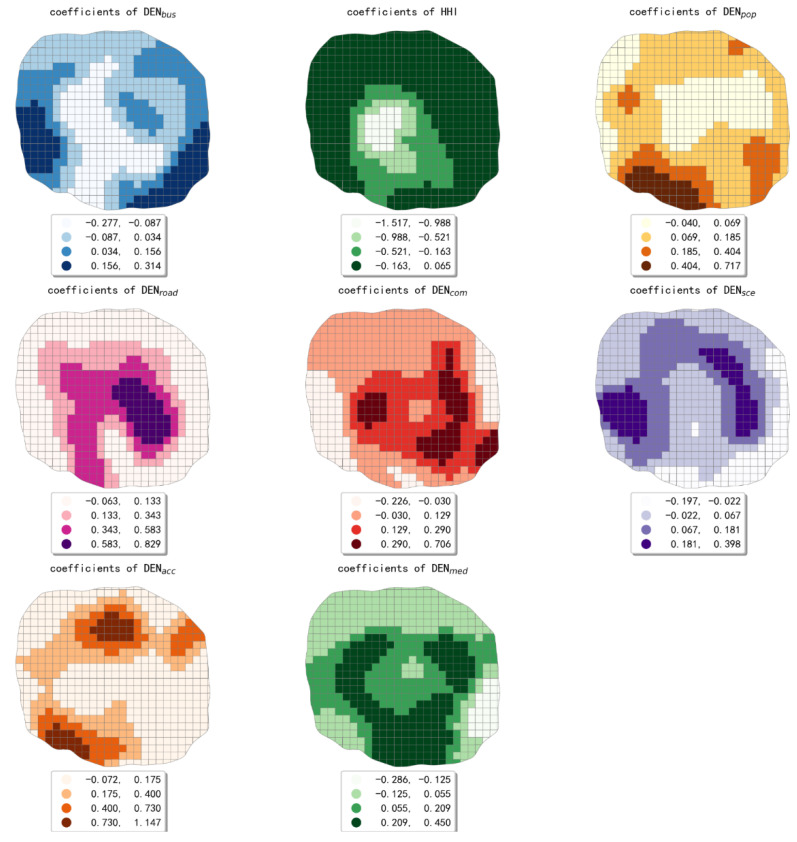
The map of coefficients estimated by the GWR model.

**Table 1 ijerph-19-16962-t001:** Data Source.

Data	Source	Format	Resolution	Time
Taxis GPS data	Intelligent China Cup (ICC), 2016	txt	/	3 August 2014, 4 August 2014
POI data	Amap	csv	/	2015
Population data	WorldPop	tiff	100 m	2014
Road data	Road traffic monitoring platform of Chengdu	geojson	/	17 July 2020
Administrative boundary	Road traffic monitoring platform of Chengdu	geojson	/	17 July 2020

**Table 2 ijerph-19-16962-t002:** COPERT emission model parameter table [46,47].

Emission Factor	CO	NOx	HC	CO_2_ *
$a_{k}$	5.497 × 10^−12^	3.856 × 10^−5^	3.549 × 10^−6^	3.32 × 10^−1^
$b_{k}$	−3.342 × 10^−2^	−8.580 × 10^−3^	−1.393 × 10^−4^	−1.76 × 10
$c_{k}$	5.110	5.773 × 10^−1^	4.738 × 10^−2^	1.45 × 10^3^
$d_{k}$	−1.044 × 10^−7^	1.307 × 10^−12^	−9.098 × 10^−14^	1.76 × 10^−11^
$e_{k}$	1.872 × 10^−3^	2.702 × 10^−18^	−6.442 × 10^−15^	8.01 × 10^−4^
$f_{k}$	−5.288 × 10^−1^	−1.308 × 10^−13^	7.726 × 10^−13^	9.13 × 10^−2^
$g_{k}$	3.751 × 10	5.431	4.015	3.51

* The model parameters for the CO_2_ emission factor were obtained by transforming the energy consumption factor (MJ·km^−1^) with a conversion factor of 69.3 g CO_2_·MJ^−1^.

**Table 3 ijerph-19-16962-t003:** The indicators of urban built environment based on “5Ds” principle.

Features	Indicators	Abbreviations
density	population density	DEN_pop_
diversity	land-use mixture	HHI
design	road density	DEN_road_
distance to transit	bus-stop density	DEN_bus_
different land use types	different POIs density	Catering POIs density: DEN_cat_ Scenic spot POIs density: DEN_sce_ Public service POIs density: DEN_pub_ Company POIs density: DEN_com_ Shopping POIs density: DEN_sho_ Transportation POIs density: DEN_tra_ Financial POIs density: DEN_fin_ Educational, scientific, and cultural POIs density: DEN_edu_ Residential district POIs density: DEN_res_ Living service POIs density: DEN_liv_ Sports and leisure POIs density: DEN_spo_ Medical service POIs density: DEN_med_ Government agency POIs density: DEN_gov_ Accommodation service POIs density: DEN_acc_

**Table 4 ijerph-19-16962-t004:** The OLS regression result of independent variables.

Variable	Coefficient	Std.Error	t-Statistic	Probability
CONSTANT	0.000	0.024	0.000	1.000
DEN_bus_	0.080	0.037	2.166	0.031 **
HHI	−0.080	0.030	−2.676	0.008 *
DEN_pop_	0.167	0.028	6.079	0.000 *
DEN_road_	0.154	0.028	5.547	0.000 *
DEN_com_	0.147	0.029	5.036	0.000 *
DEN_sce_	0.116	0.028	4.138	0.000 *
DEN_acc_	0.228	0.030	7.662	0.000 *
DEN_med_	0.262	0.033	7.992	0.000 *

* Significant in the 0.01 level; ** significant in the 0.05 level.

**Table 5 ijerph-19-16962-t005:** The diagnostic information of two models.

Model	Adj.R2	RSS	AICc	Moran’s I of Residual	Z normal	*p* Value
OLS	0.673	189.551	1023.397	0.282	13.758	0.000 *
GWR	0.854	67.677	711.946	−0.004	−0.118	0.905

* Significant in the 0.01 level.

**Table 6 ijerph-19-16962-t006:** Summary statistics for GWR parameter estimates.

Variable	Mean	STD	Min	Median	Max
DEN_bus_	0.023	0.134	−0.277	0.016	0.314
HHI	−0.167	0.299	−1.517	−0.026	0.065
DEN_pop_	0.140	0.142	−0.040	0.101	0.717
DEN_road_	0.214	0.222	−0.063	0.125	0.829
DEN_com_	0.105	0.156	−0.226	0.081	0.706
DEN_sce_	0.060	0.096	−0.197	0.039	0.398
DEN_acc_	0.244	0.244	−0.072	0.155	1.147
DEN_med_	0.099	0.156	−0.286	0.101	0.450

## Data Availability

The taxis trajectory data used to support this study are provided by the Intelligent China Cup, 2016. Due to the security policies, the authors have no right to disclose, publish, copy, or conduct all original datasets. Readers in need, please visit the URL (https://challenge.datacastle.cn/v3/cmptDetail.html?id=175, accessed on 12 November 2016) for further information.

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
