# Peer review of "Marginal Effects and Spatial Variations of the Impact of the Built Environment on Taxis’ Pollutant Emissions in Chengdu, China"

_ijerph, 2022, doi:10.3390/ijerph192416962_

Round 1
Reviewer 1 Report
Comments to the Author
Manuscript ID: ijerph-2059100
The author presented the marginal effects and spatial heterogeneity of built environment’s effects on taxicab traffic emissions in Chengdu, China. The paper is well written and appropriately organized. However, some issues should be considered before it is accepted for publication.
(1) Sample size is quite small, only two days: 3 Aug 2014 (Sunday, weekend) and 4 Aug 2014 (Monday, Weekend). Therefore a conclusion “The result showed that the daily taxis emission on weekday was higher than emission on weekend” in abstract is not meaningful enough. Why did author choose these days, and only two days is used?
(2) Line 318-319: “We speculated that the reason may be mainly due to the more random travel behavior and higher average taxi travel speed during weekend.” This statement is not clear enough. What is “random travel behavior”?
(3) Time interval of data sources is not consistent as shown in Table 1. It is difficult to provide a general conclusion if we based on these data. In addition, time of taxi GPS data is both 3rd and 4th Aug 2014.
(4) Figure 3: information on sampling days should be provided (e.g., 3rd and 4th Aug 2014)
(5) Comparison to previous studies is weak. Please consider all Section 4 (results and discussion). For example, “The most significant inhibitory effect was found in the range of 16000 people/Km2 to 22000 people/Km2”, this result is consistent with other places or not? You should also look other results and revise them.
Reviewer 2 Report
1. The title suggests that the research concerns the impact of the built environment on CO2 emissions, while the text also mentions other pollutants (e.g., Fig.5)
2. The article does not refer to the Air Quality Limits coefficient for China
2. In the reviewer's opinion, the presentation of the impact of spatial differentiation in figures 3 to 7 needs to be revised. It is not known what GIS program was used to generate these maps.
3. Spatial analyzes concerning OLS and GWR could be performed based on ArcGis (regression tools).
4. Figures 3 and 4 lack information on impurities (Vertical legend and figure caption)

Round 2
Reviewer 2 Report
I can now accept the article as my comments have been taken into account.